# Cost-Effectiveness of Screening Algorithms for Familial Hypercholesterolaemia in Primary Care

**DOI:** 10.3390/jpm12030330

**Published:** 2022-02-22

**Authors:** Matthew Jones, Ralph K. Akyea, Katherine Payne, Steve E. Humphries, Hasidah Abdul-Hamid, Stephen Weng, Nadeem Qureshi

**Affiliations:** 1NIHR School for Primary Care Research, University of Nottingham, Nottingham NG7 2UH, UK; ralph.akyea1@nottingham.ac.uk (R.K.A.); or hasidah@uitm.edu.my (H.A.-H.); stephen.weng@evda.co.uk (S.W.); nadeem.qureshi@nottingham.ac.uk (N.Q.); 2Manchester Centre for Health Economics, School of Health Sciences, The University of Manchester, Manchester M13 9PL, UK; katherine.payne@manchester.ac.uk; 3Institute of Cardiovascular Science, University College London, London WC1E 6HX, UK; steve.humphries@ucl.ac.uk; 4Department of Primary Care Medicine, Faculty of Medicine, Jalan Hospital, Universiti Teknologi MARA, Sungai Buloh 47000, Malaysia

**Keywords:** economic evaluation, cost-effectiveness, genetics, electronic health records, familial hypercholesterolemia

## Abstract

Although familial hypercholesterolemia (FH) screening within primary care is considered cost-effective, which screening approach is cost-effective has not been established. This study determines the cost-effectiveness of six case-finding strategies for screening of electronic health records to identify index patients who have genetically confirmed monogenic FH in English primary care. A decision tree was constructed to represent pathways of care for each approach (FH Case Identification Tool (FAMCAT) versions 1 and 2, cholesterol screening, Dutch Lipid Clinic Network (DLCN), Simon Broome criteria, no active screening). Clinical effectiveness was measured as the number of monogenic FH cases identified. Healthcare costs for each algorithm were evaluated from an NHS England perspective over a 12 week time horizon. The primary outcome was the incremental cost per additional monogenic FH case identified (ICER). FAMCAT2 was found to dominate (cheaper and more effective) cholesterol and FAMCAT1 algorithms, and extendedly dominate DLCN. The ICER for FAMCAT2 vs. no active screening was 8111 GBP (95% CI: 4088 to 14,865), and for Simon Broome vs. FAMCAT2 was 74,059 GBP (95% CI: −1,113,172 to 1,697,142). Simon Broome found the largest number of FH cases yet required 102 genetic tests to identify one FH patient. FAMCAT2 identified fewer, but only required 23 genetic tests.

## 1. Introduction

Familial hypercholesterolaemia (FH) is an autosomal dominant disorder, caused by carrying a mutation in any of four genes (*LDLR*/*APOB*/*PCSK9*/*APOE*) [1]. Individuals with FH have elevated blood concentrations of low-density lipoprotein cholesterol (LDL-C) since birth and, if untreated, are at higher risk of developing early cardiovascular disease than the normal population [2,3]. Treatment with high-intensity statins is recommended by NICE (CG71) [4] and has been shown to substantially reduce both morbidity [5] and mortality [6]. In the UK, around one in 250 individuals are thought to have FH [7], of whom fewer than 10% have currently been identified [8].

Decision making in various countries requires that new healthcare technologies demonstrate value for money [9]. While screening for FH within primary care has been demonstrated to be cost-effective [10], there are varying approaches with little evidence as to which offers best value for money [4,11]. Furthermore, there has been recent development of various algorithms which can screen a patient’s electronic health record (EHR) and assess their risk of having FH [12].

The aim of this study was to determine the cost-effectiveness of five approaches for the systematic screening of EHRs to identify index patients who have genetically confirmed monogenic FH registered to typical English primary care practice, compared with no active screening of EHRs.

## 2. Materials and Methods

An early decision-analytic model was conceptualised and built to address the decision problem (see Appendix A) following published recommendations [13,14,15], and reported in line with published criteria [16].

### 2.1. Population

A typical primary care practice in NHS England would have approximately 10,000 patients registered, with 4500 patients having a cholesterol measurement recorded and being eligible for screening of EHR to detect possible FH. The hypothetical cohort was 4500 adult patients, with a mean age of 56 years. This profile of patients represents those who took part in the FH Case Identification Tool (FAMCAT) feasibility study [17], funded by the National Institute of Health Research (NIHR) School for Primary Care Research (Grant SPCR 332).

### 2.2. Interventions and Comparator

Table 1 describes the five interventions potentially possible given the availability of algorithms and case-finding criteria that can be used to guide how to screen the EHR of patients in primary care. The comparator was chosen because, although screening within primary care is recommended [4], expert consensus from six clinical members of the FAMCAT feasibility study steering committee suggested that there is little evidence that any form of FH screening is actually carried out in primary care. Three case-finding criteria were selected because they represent the current recommended approaches across several countries (cholesterol, Dutch Lipid Clinic Network (DLCN), and Simon Broome (SB)) [4,18], along with two newly developed algorithms which have demonstrated increased sensitivity and specificity from database studies (FAMCAT 1 and 2) [12,19].

### 2.3. Study Design

We determined the possible patient pathways using expert consensus, formed from eight clinical members of the FAMCAT feasibility study steering committee. A decision tree was selected as the most appropriate decision analytic model for the selected time horizon of the evaluation [20]. It was recommended by clinical experts that the focus should be on three types of patients: those with a positive monogenic FH mutation, those with a variant of unknown significance (VUS), and those with no relevant mutation. The model’s first stage split the three types of patients before engaging with the primary care (see nodes A and C in Figure 1), and then determined whether the patient has fulfilled the diagnostic criteria or not (see nodes B, D, and E in Figure 1).

For patients in the comparator arm, as no screening was conducted, the pathway involved no primary care involvement. The initial pathway for all the other interventions was initial screening of electronic health record, requesting the family history questionnaire and updating the electronic health record, and finally secondary screening of electronic record. For those who did not fulfil the diagnostic criteria for the interventions, it was assumed that they would receive no further treatment. Patients who fulfil the diagnostic criteria are invited to primary care to take a blood sample for genetic testing, with the genetic test done at a secondary care facility. Once the patient’s general practitioner (GP) received the result of the genetic test, if a patient was found to have a FH mutation or a VUS, then the patient was required to attend a double consultation with their GP to inform them of their results. If the genetic test detected no relevant mutation, then the patient was informed of the result via a letter from their GP.

### 2.4. Model Inputs

The decision tree required three types of model inputs: probabilities, costs, and outcomes (see Appendix A). According to the estimates used for the proportions, in the cohort of 4500 adult patients eligible for screening, 16 would have monogenic FH, nine would have a VUS, and the remaining 4475 would have no relevant mutation.

### 2.5. Costs

As the pathway for the comparator assumed no contact with primary care services, the cost per patient was 0 GBP. For the intervention pathway, healthcare resource use was based on estimates from the FAMCAT feasibility study. The intervention pathways were split into three parts, and the total costs per patient for each part of the pathway can be found in Appendix A. For the first part of the intervention pathways, resource use included practice administrative time for performing EHR searches, letter preparation, posting, and entering the family history questionnaire, as well as time spent by the GP on reviewing the results of the searches. Resource use for the first part was estimated across all study practices, so the per patient cost was estimated by dividing the total cost for the searches by number of eligible patients registered (86,219 patients).

For the second part of the intervention pathways, resource use focused on performing blood tests within primary care, posting samples, and analysing samples for the genetic test in secondary care. These resources were only used by patients who fulfilled the diagnostic criteria and were applied on a per patient level. For the third part of the intervention pathways, resource use included posting letters to patients, and, for those who had a FH mutation or VUS, a double consultation to discuss the genetic test results with their GP.

### 2.6. Analytical Strategy

We used the standard method for evaluating multiple technologies [21], details of which can be found in Appendix A.

### 2.7. Sensitivity Analyses

A probabilistic sensitivity analysis (PSA) whereby all model inputs could vary simultaneously was performed via 1000 Monte Carlo simulations using standard techniques [22], details of which can be found in Appendix A. Information on uncertainty was taken from the literature, using the FAMCAT feasibility study data by estimating Wilson 95% confidence intervals (CI) [23], or assuming a standard error of 10% of the mean value if no information was identified [22]. Results of the PSA were plotted as scatterplots on the incremental cost-effectiveness plane and cost-effectiveness acceptability curves (CEACs) for each intervention. A one-way sensitivity analysis was performed on the proportion of patients with no relevant mutation being identified as above risk (Node E in Figure 1), with values varied between zero and twice the mean value for each of the algorithms simultaneously, in 10% increments. Two scenario analyses were performed on the diagnostic thresholds for the FAMCAT1 and FAMCAT2 algorithms, simultaneously changing the proportion of patients identified by each algorithm for nodes B, D, and E (see Appendix A). A final scenario analysis was performed on the proportion of patients with VUS within the cohort, using a value of three in 100 for the proportion of patients with monogenic FH [24].

### 2.8. Patient and Public Involvement

Patient representatives participated in developing the model and interpreting the study results.

## 3. Results

Estimated expected costs and outcomes are reported in Table 2. After ranking the interventions by cost, the incremental analysis suggested that FAMCAT2 dominated FAMCAT1 and cholesterol, while extendedly dominating DLCN. This was because FAMCAT2 required the fewest number of genetic tests to identify one monogenic FH case, with the cost of the genetic test being the largest component of the costs of the different interventions. However, FAMCAT2 did not dominate SB, since SB yielded the greatest number of FH cases identified but at a much higher expected total cost per patient. This was because the SB criteria required a large number of genetic tests to find one monogenic FH case.

### 3.1. Probabilistic Sensitivity Analysis

PSA results can be found in Table 3, while the scatterplot for the incremental analysis can be found in Figure 2A. The PSA supported the initial findings, and there was no change in the ranking of the interventions, with FAMCAT2 appearing to dominate FAMCAT1 and cholesterol screening, and extendedly dominating DLCN. However, the finding was nonsignificant as the 95% CI for incremental FH cases identified crossed zero, implying there were scenarios where DLCN, FAMCAT1, and cholesterol were preferred over FAMCAT2. The CEACs in Figure 2B demonstrated that, at most, FAMCAT1 had a 34% chance of being preferred to DLCN, while cholesterol had a chance of 24% of being preferred to FAMCAT2. FAMCAT2 had at least a 30% chance of being preferred to DLCN, increasing to 97% chance at a higher willingness to pay per monogenic FH case identified. SB continued not to be dominated, although there were scenarios where it was estimated that SB identified fewer monogenic FH cases than FAMCAT2.

### 3.2. One-Way Sensitivity Analysis

Results of the sensitivity analysis for the algorithm identifying a patient with no relevant mutation as above risk can be found in Appendix A. Overall results were similar to the initial findings apart from the analysis where the proportion of patients with no relevant mutation being identified as above risk was zero. In that scenario, SB extendedly dominated all other interventions, with an incremental cost-effectiveness ratio (ICER) of 1105 GBP per monogenic FH case identified when compared to no active screening. The analysis demonstrated that the ICER for SB was particularly sensitive to changes in the investigated proportion, while the ICER for FAMCAT2 was less sensitive.

### 3.3. Scenario Analysis

Findings for the scenario analysis of changing FAMCAT1 and FAMCAT2 thresholds can be found in Appendix A. At thresholds of 0.050 or higher, FAMCAT1 was either dominated or extendedly dominated by the FAMCAT2 algorithm or DLCN criteria. For thresholds at or below 0.010, FAMCAT1 was no longer dominated by any intervention and did identify all cases of monogenic FH within the cohort. However, it required a large number of genetic tests to identify one monogenic FH case and, consequently, was very expensive, with corresponding large ICERs.

At a threshold of 0.002, FAMCAT2 identified most but not all of the FH cases within the cohort, but required a large number of genetic tests and, thus, was very expensive and extendedly dominated by SB. For thresholds between 0.0036 and 0.0047, the result of the evaluation was similar or identical to the initial findings. At a threshold of 0.010, FAMCAT2 identified 50% of monogenic FH cases while only requiring six genetic tests to identify one FH patient, thus having a very low expected cost per patient. At this threshold, the algorithm was dominant over all other interventions except for SB. At a threshold of 0.050 and above, FAMCAT2 now identified very few cases of FH at a very low cost. FAMCAT2 no longer dominated or extendedly dominated any other intervention. In this scenario, the ranking was no active screening, FAMCAT2, DLCN, cholesterol, and SB, with FAMCAT1 extendedly dominated by Dutch Lipid.

Estimates using the three in 100 value of monogenic FH prevalence for the prevalence of VUS can be found in Appendix A. Other than a very slight increase in expected total costs and ICERs, the change in prevalence of VUS did not impact the initial findings. FAMCAT2 continued to dominate the FAMCAT1 algorithm and cholesterol criteria, while it extendedly dominated DLCN, but not SB.

## 4. Discussion

We investigated the cost-effectiveness of potential case-finding criteria for identifying monogenic FH in primary care. Our findings suggest that two approaches, the FAMCAT2 algorithm and SB criteria, were preferable over the other three interventions. FAMCAT2 identified most cases of monogenic FH within our cohort, requiring the fewest number of genetic tests to find one monogenic FH case, and consequently having favourable ICERs. SB identified the most cases of monogenic FH; however, it required a much larger number of genetic tests to find one case and, thus, a large, expected cost per patient compared to the other interventions, with a corresponding large ICER when compared with FAMCAT2.

### 4.1. Strengths

This analysis was based on current relevant costs to the NHS of the note searching, clinician, and healthcare professional time, genetic and biochemical testing, and management costs. As such, it gives an accurate estimate of actual costs likely to be incurred in the use of these different algorithms in general practice in the UK. The modelling was based on data obtained from a representative group of general practices and included note searching from over 86,219 individuals. As such, it is likely to be directly relevant to methods for FH case finding in general practice in the UK as recommended by NICE guidelines [4].

### 4.2. Limitations

We only included a short time horizon, omitting potential long-term consequences that may occur from the treatment of patients with FH. FH is associated with long-term outcomes, such as premature heart attacks and heart disease, all of which can have a serious impact on the health and quality of life. Furthermore, treatment is relatively successful, and can mitigate most of the impacts of FH [6,25]. The omission of the impacts of FH and the benefits of successful treatment suggests that we are conservatively estimating the cost-effectiveness of the different screening algorithms. To model long-term cost-effectiveness would require additional data on quality of life, impact of treatment, and uptake of treatment, none of which were captured as part of the original study [12,17]. However, it has been estimated that over 80% of the lifetime treatment costs associated with FH are accrued within the first year of diagnosis [26]; therefore, there would be little impact on healthcare resources beyond a time horizon of 1 year.

This evaluation does not include any benefits or cases of FH identified through cascade testing. Cascade testing is important since FH is a monogenic disorder; hence, once one family member has been identified as having FH, all first-degree relatives are at 50% chance of also having FH and can benefit from receiving appropriate treatments. Cascade testing has been shown to be both feasible, acceptable, and cost-effective [10,26,27,28]; therefore, it can be argued that the omission of the benefits from cascade testing will give conservative estimates for cost-effectiveness. However, the impact of cascade testing was considered out of scope for our evaluation, since the aim of the study was to identify the cost-effectiveness of index case identification. Since cascade testing has already been shown as cost-effective, then, once we have identified a cost-effective way of identifying index cases, we already know that cascade testing should be offered.

We assumed a worst-case scenario for intervention costs, in that all patients who are identified as being above risk consent to and complete a genetic test and attend any subsequent GP consultations if required. This could be considered unrealistic, as previous studies have shown that only 43% of patients with possible FH actually attended a GP consultation when informed they may be of risk of FH [29]. By using a worst-case scenario for resource use, we could be over-inflating the costs associated with each screening algorithm and potentially over-estimating the number of patients who are confirmed with FH. However, it is likely that loss to follow-up would be the same irrespective of which algorithm was used; therefore, the use of the worst-case scenario is unlikely to change the overall ranking of the algorithms.

The cohort study used specifically focuses on patients that already have a high recorded cholesterol level to inform the model inputs for the effectiveness of the different screening algorithms. This creates two issues for the evaluation. Firstly, we may not be estimating the true prevalence of monogenic FH within the population, since the algorithms only focus on those which report clinical features of FH which is a smaller population of those with monogenic FH. Secondly, since the study was non-random, there may be biases within the data that could influence our results. The group of patients was restricted to those with high cholesterol; thus, the interventions may have overestimated the number of patients who do not have a relevant mutation as being flagged as above risk by the case-finding criteria. Had the interventions been applied to a sample more representative of the general primary care population, the proportion of individuals with no relevant mutation being flagged as above risk may have been lower, reducing the number of genetic tests and the total expected cost per patient for each intervention. However, as demonstrated in the one-way sensitivity analysis, the overall finding of the FAMCAT2 algorithm and SB criteria being preferred over the other interventions did not change, suggesting that this may not have influence over the evaluation’s findings.

### 4.3. In Context with Other Work

There are no known studies that evaluated different electronic screening criteria for index case identification of FH within primary care, preventing direct comparison. Crosland et al. evaluated several strategies for diagnosing FH within primary care amongst cascade patients [10], which included using both the DLCN and the SB criteria for index case identification, as well as a strategy of using a genetic test on its own. All three strategies were found to be cost-effective, and well below commonly accepted UK decision-making thresholds [30]. We also found the DLCN and SB criteria to be cost-effective; however, DLCN was outperformed by the FAMCAT2 algorithm. Therefore, it is likely that, had the FAMCAT2 algorithm been available at the time when Crosland et al. were conducting their evaluation, it too would have also been considered cost-effective.

### 4.4. Implications for Policy

Our findings suggest that using electronic criteria to screen patients’ electronic health records is a highly cost-effective approach for identifying index cases of FH within primary care. The use of the SB criteria yielded the most cases identified but required a large expenditure of healthcare resources. In contrast, while FAMCAT2 did not identify as many cases as SB, it was considerably cheaper and a good combination of sensitivity and specificity. While the other approaches had utility, they were dominated by FAMCAT2 as they required more resources to detect fewer or the same number of cases of FH.

## Figures and Tables

**Figure 1 jpm-12-00330-f001:**
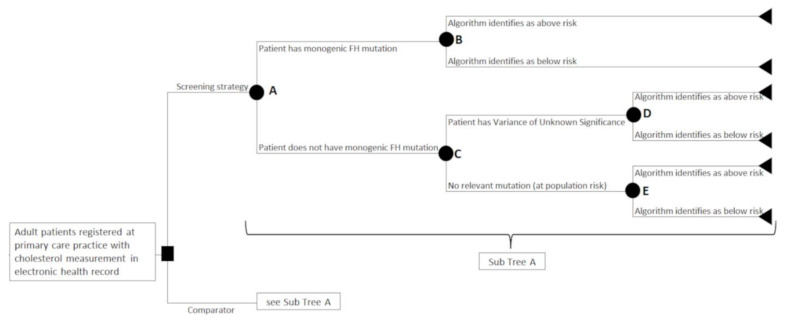
Decision tree structure for all interventions, with squares denoting decision nodes, circles denoting chance nodes, and triangles representing the end of the tree.

**Figure 2 jpm-12-00330-f002:**
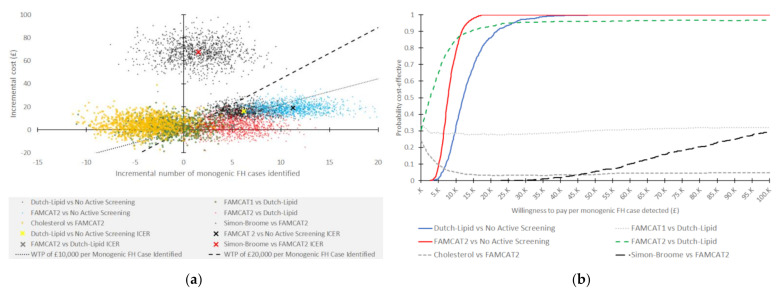
Results of the probabilistic sensitivity analysis: (**a**) scatterplot of incremental costs and incremental cases detected on the cost-effectiveness plane for each comparison; (**b**) cost-effectiveness acceptability curves for each comparison.

**Table 1 jpm-12-00330-t001:** Description of interventions and comparator.

Name of Intervention	Action	Requirements to Run Case-Finding Criteria
No active, systematic screening of electronic health records	Patient electronic health records are not screened for possible markers of FH	None
Cholesterol ^1^ [4]	Search electronic health records for individuals who are either (i) younger than 30 years old with a total cholesterol concentration greater than 7.5 mmol/L, or (ii) 30 years or older with a total cholesterol concentration greater than 9.0 mmol/L. Current approach as recommended by NICE [4]	Most recent LDL-cholesterol concentration level
Dutch Lipid Clinic Network ^1^ [4]	Points-based criteria. Points awarded on the basis of symptoms, cholesterol levels, family history of illness, and/or DNA test. Patients are scored, with a score of eight or greater having definite FH, and a score of five or greater as possible FH.	Untreated LCL-C recorded, family history of premature coronary and/or vascular disease, first-degree relative with known LDL-cholesterol above 95th percentile, tendinous xanthomata and/or arcus cornealis, clinical history of premature coronary artery disease, cerebral, or peripheral vascular disease
Simon Broome Criteria ^1^ [4]	Category-based criteria based on a patient’s cholesterol levels, family history of premature CHD or high cholesterol, and/or DNA test. Patients are either coded as definite or probable FH.	Age, total cholesterol, LDL cholesterol, tendon xanthomas in patient, first- or second-degree relative, DNA-based evidence of a functional LDLR, PCSK9, and APOB mutation, family history of premature CVD events, family history of extremely high cholesterol.
Familial Hypercholesterolaemia Case Identification Tool version 1 (FAMCAT1) ^1^ [12]	A multivariate logistic regression model, consisting of nine diagnostic indicators stratified by gender. Age, cholesterol levels, and triglycerides are categorised. Algorithm identifies patients at increased risk of FH.	Gender, total cholesterol or LDL-cholesterol, age during cholesterol measurement, triglycerides, lipid-lowering drug usage, family history of FH, family history of CHD, family history of raised cholesterol, diabetes, and chronic kidney disease.
Familial Hypercholesterolaemia Case Identification Tool version 2 (FAMCAT2) ^1^ [12]	An updated FAMCAT1 algorithm, with re-estimated regression equations using continuous variables for total cholesterol, LDL-cholesterol, triglycerides, and age. Algorithm identifies patients at increased risk of FH.	As above

^1^ Requires a search of electronic health records.

**Table 2 jpm-12-00330-t002:** Initial findings for the evaluation, with algorithms ranked by expected total cost per patient.

Intervention	Expected Total Cost per Patient (GBP; 2018/2019)	Number of FH Cases Identified	Number of Genetic Tests to Fine One FH Case	Incremental Cost (GBP) ^1^	Incremental Number of FH Cases Identified	Incremental Cost per Additional FH Case Identified (GBP; 2018/2019)	Notes
No active screening	£0	0	-	-	-	-	
Dutch Lipid	£16.12	6	35	16.12	6.18	11,734	Versus no active screening, extendedly dominated ^3^ by FAMCAT 2
FAMCAT1	£18.51	5	49	2.39	−1.03	Dominated ^2^	Dominated by Dutch Lipid
FAMCAT2	£19.02	11	23	19.02	11.33	7552	Versus no active screening
Cholesterol	£23.63	7	46	4.61	−4.12	Dominated ^2^	Dominated by FAMCAT2
Simon Broome	£87.28	13	102	68.26	1.49	206,431	Versus FAMCAT 2

^1^ Defined as expected total cost per patient for intervention minus expected total cost per patient for comparator. ^2^ Occurs when an intervention is more expensive and less effective than the comparator. ^3^ Occurs when the ICER is higher than the next more effective alternative.

**Table 3 jpm-12-00330-t003:** Results for the probabilistic sensitivity analysis, with algorithms ranked by mean expected total cost per patient.

Screening Algorithm	Expected Total Cost per Patient (GBP) Mean (95% CI)	Number of FH Cases Identified Mean (95% CI)	Number of Genetic Tests Required to Identify One Patient with Monogenic FH Mean (95% CI)	Incremental Cost (GBP) Mean (95% CI)	Incremental Number of FH Cases Identified Mean (95% CI)	Incremental Cost per Additional FH Case Identified (GBP) Mean (95% CI)
No active screening	0 (0-0)	0 (0-0)	0 (0-0)	-	-	-
Dutch Lipid	16.32 (9.16–25.80)	6 (3–11)	40 (14–93)	16.32 (9.16–25.80) ^1^	6 (3–11) ^1^	13,528 (5395–31,086) ^1^
FAMCAT1	19.00 (10.91–29.02)	5 (2–9)	60 (22–150)	2.69 (−9.30–15.01) ^2^	−1 (−6–4) ^2^	2946 (−138,736–123,755) ^2^
FAMCAT2	19.23 (11.59–28.75)	11 (7–17)	24 (11–46)	19.23 (11.59–28.75) ^1^	11 (7–17) ^1^	8111 (4088–14,865) ^1^
				2.91 (−8.91–14.52) ^2^	5 (0–10) ^2^	6118 (−22,023–32,018) ^2^
Cholesterol	23.83 (15.37–34.32)	7 (4–12)	51 (23–103)	4.61 (−7.82–17.44) ^3^	−4 (−9–1) ^3^	−16,589 (−56,929–50,116) ^3^
Simon Broome	86.92 (70.95–104.80)	13 (7–19)	109 (66–188)	67.70 (48.47–87.22) ^3^	2 (−5–7) ^3^	74,059 (−1,113,172–1,697,142) ^3^

^1^ Incremental compared to no active screening. ^2^ Incremental compared to Dutch Lipid. ^3^ Incremental compared to FAMCAT2.

## Data Availability

The data presented in this study are available in Appendix A.

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
