# Peer review of "Cost-Effectiveness of Screening Algorithms for Familial Hypercholesterolaemia in Primary Care"

_jpm, 2022, doi:10.3390/jpm12030330_

Round 1

Reviewer 1 Report

The authors have performed a comprehensive study here. The length of the study period needs to be increased for better comparison between screening techniques. Moreover, suspected cases based on reported clinical features were only tested for mutations, missing the actual prevalence. Although, the authors have shown that FAMCAT2 is cost-effective, minimum genetic screening is essential. 

Author Response

  • The authors have performed a comprehensive study here.

We thank the reviewer for their positive review and helpful comments.

  • The length of the study period needs to be increased for better comparison between screening techniques.

This is an early economic analysis that takes the chosen perspective to produce indicative estimates of the costs and consequences of different screening approaches for monogenic FH within primary care. A longer time horizon requires additional data sources. We have highlighted on pages 8 to 9, lines 241 to 252 the limitations for taking the assumed time horizon and what additional data would be needed to conduct the analysis from a longer time horizon.

  • Moreover, suspected cases based on reported clinical features were only tested for mutations, missing the actual prevalence.

This is a useful comment. We have added a short paragraph to the discussion highlighting this limitation, please see page 9 lines 272 to 275.

  • Although, the authors have shown that FAMCAT2 is cost-effective, minimum genetic screening is essential. 

This study has demonstrated that because genetic testing is expensive it is necessary to target the effective use of health care resources. This study suggests that by using cost-effective screening algorithms, it is possible to target genetic testing so that the cost burden to the National Health Service is not excessive given the potential benefits to patients. We have not made any amendments to the manuscript.

Reviewer 2 Report

There are no outstanding problems with the scientific research presented in this manuscript but there might be a few ways to improve how it is presented.

Table 1: Showing this table in one single page can help to understand it better.

Figure 1: The font used here its extremely small and hence difficult to read. Also, since all six case finding strategies are followed by the same “Sub Tree A” there is no need to list all of them. It might be possible to just mention that all of them are preceded and followed by the same node/sub tree.

Appendix 2 contains information that it is vital to understand the whole manuscript hence some of its information should be moved to the main text.

Table 2: Since FAMCAT2 appears to be the overall  best strategy it might be better to use it as the baseline against which all the others are compared instead of having of keep track of which strategy dominates which one.

Since many of the numbers in Table 2 appear as the means in Table 3, e.g. all the numbers in the column “Number of FH cases identified” of Table 2 appear again as the means in this same column in Table 3, maybe Tables 2 and 3 could be merged.  It should be possible to insert the whole table in one single landscape-oriented page.

Author Response

  • There are no outstanding problems with the scientific research presented in this manuscript but there might be a few ways to improve how it is presented.

We thank the reviewer for their positive review and helpful comments. We have tried our best to meet their requirements, and give a detailed response below.

  • Table 1: Showing this table in one single ccc can help to understand it better.

This comment will be picked up when the final print version of the manuscript is prepared. We will endeavour to make sure the production unit is aware of this advice. In the meantime, we have added a page break to shift Table 1 to the following page. See Page 3, line 80. We have also added page breaks to shift Tables 2 and 3 onto a single page each. See page 6, line 162 and page 7, line 180.

  • Figure 1: The font used here its extremely small and hence difficult to read. Also, since all six case finding strategies are followed by the same “Sub Tree A” there is no need to list all of them. It might be possible to just mention that all of them are preceded and followed by the same node/sub tree.

This comment will be picked up when the final print version of the manuscript is prepared. We will endeavour to make sure the production unit is aware of this advice. In the meantime, we have increased the text size in the all the figures to make things more readable, and adjusted Figure 1 such that all the interventions are no longer listed. Please see page 4 line 92. For changes in Figure 2, please see page 7 line 183.

  • Appendix 2 contains information that it is vital to understand the whole manuscript hence some of its information should be moved to the main text.

We cannot comply with this suggestion due to space requirements for publication in the journal. We have endeavoured to include the key design criteria in the main body of the manuscript but if the reviewer has some more specific suggestions as to which elements of Appendix 2 are important enough for the main text, taking into account word and figure/table count restrictions then we will amend the manuscript accordingly. We are also happy to take advice from the editor on this matter.

  • Table 2: Since FAMCAT2 appears to be the overall  best strategy it might be better to use it as the baseline against which all the others are compared instead of having of keep track of which strategy dominates which one.

This study conducted a full incremental analysis, which is the standard and correct methodological approach in  scenarios in which there are more than two interventions to compare [1]. This approach is  recommended as part of the reference case by the National Institute of Health and Care Excellence (NICE). [2]

  • Since many of the numbers in Table 2 appear as the means in Table 3, e.g. all the numbers in the column “Number of FH cases identified” of Table 2 appear again as the means in this same column in Table 3, maybe Tables 2 and 3 could be merged.  It should be possible to insert the whole table in one single landscape-oriented page.

We thank the reviewer for this suggestion; however, we haven’t made this change as Tables 2 and 3 demonstrate the results from two separate analyses. Table 2 represents the results for the initial inputs of the model, whereas Table 3 represents the results of the probabilistic sensitivity analysis whereby we allow for all uncertainty within model inputs. [1, 3, 4] Combining tables 2 and 3 would merge two separate and different analyses, and we feel this would confuse the reader.   

References

  1. Drummond, M.F., et al., Methods for the Economic Evaluation of Health Care Programmes. 3rd ed. 2005: Oxford University Press, USA.
  2. National Institute for Health and Care Excellence. Guide to the methods of technology appraisal 2013. 2013 04/05/2013 [cited 2020 19/12/2020]; Available from: https://www.nice.org.uk/process/pmg9/chapter/the-reference-case.
  3. Briggs, A., M.J. Sculpher, and K. Claxton, Decision Modelling for Health Economic Evaluation. 1st ed. Handbooks for Health Economic Evaluation. 2006: Oxford University Press, USA.
  4. Claxton, K., et al., Probabilistic sensitivity analysis for NICE technology assessment: not an optional extra. Health Econ, 2005. 14(4): p. 339-47.